# Epigenetic Signatures of Smoking in Five Brain Regions

**DOI:** 10.3390/jpm12040566

**Published:** 2022-04-02

**Authors:** Lea Zillich, Eric Poisel, Fabian Streit, Josef Frank, Gabriel R. Fries, Jerome C. Foo, Marion M. Friske, Lea Sirignano, Anita C. Hansson, Markus M. Nöthen, Stephanie H. Witt, Consuelo Walss-Bass, Rainer Spanagel, Marcella Rietschel

**Affiliations:** 1Department of Genetic Epidemiology in Psychiatry, Central Institute of Mental Health, Medical Faculty Mannheim, Heidelberg University, 68159 Mannheim, Germany; lea.zillich@zi-mannheim.de (L.Z.); eric.poisel@zi-mannheim.de (E.P.); fabian.streit@zi-mannheim.de (F.S.); josef.frank@zi-mannheim.de (J.F.); jerome.foo@zi-mannheim.de (J.C.F.); lea.sirignano@zi-mannheim.de (L.S.); stephanie.witt@zi-mannheim.de (S.H.W.); marcella.rietschel@zi-mannheim.de (M.R.); 2Louis A. Faillace, MD, Department of Psychiatry and Behavioral Sciences, McGovern Medical School, University of Texas Health Science Center at Houston, Houston, TX 77054, USA; gabriel.r.fries@uth.tmc.edu (G.R.F.); consuelo.walssbass@uth.tmc.edu (C.W.-B.); 3Institute of Psychopharmacology, Central Institute of Mental Health, Medical Faculty Mannheim, University of Heidelberg, 68159 Mannheim, Germany; marion.friske@zi-mannheim.de (M.M.F.); anita.hansson@zi-mannheim.de (A.C.H.); 4Institute of Human Genetics, School of Medicine & University Hospital Bonn, University of Bonn, 53127 Bonn, Germany; markus.noethen@uni-bonn.de; 5Center for Innovative Psychiatric and Psychotherapeutic Research, Biobank, Central Institute of Mental Health, Medical Faculty Mannheim, Heidelberg University, 68159 Mannheim, Germany

**Keywords:** DNA methylation, tobacco smoking, neurocircuitry, EWAS, addiction

## Abstract

(1) Background: Epigenome-wide association studies (EWAS) in peripheral blood have repeatedly found associations between tobacco smoking and aberrant DNA methylation (DNAm), but little is known about DNAm signatures of smoking in the human brain, which may contribute to the pathophysiology of addictive behavior observed in chronic smokers. (2) Methods: We investigated the similarity of DNAm signatures in matched blood and postmortem brain samples (*n* = 10). In addition, we performed EWASs in five brain regions belonging to the neurocircuitry of addiction: anterior cingulate cortex (ACC), Brodmann Area 9, caudate nucleus, putamen, and ventral striatum (*n* = 38–72). (3) Results: cg15925993 within the *LOC339975* gene was epigenome-wide significant in the ACC. Of 16 identified differentially methylated regions, two (*PRSS50* and *LINC00612/A2M-AS1*) overlapped between multiple brain regions. Functional enrichment was detected for biological processes related to neuronal development, inflammatory signaling and immune cell migration. Additionally, our results indicate the association of the well-known *AHRR* CpG site cg05575921 with smoking in the brain. (4) Conclusion: The present study provides further evidence of the strong relationship between aberrant DNAm and smoking.

## 1. Introduction

Tobacco smoking has a strong impact on human health and has been identified as a risk factor for a variety of diseases [1,2]. While the mechanisms are still unclear, it has been assumed that epigenetic factors, especially DNA methylation (DNAm) changes, play a role in the pathophysiology of smoking-associated diseases [3,4]. To date, most epigenetic association studies (EWASs) in the context of smoking have been conducted in peripheral blood, largely given the convenient, less-invasive sampling procedure in comparison to other tissues. The largest EWAS meta-analysis of smoking in blood (*n* = 15,907) found 18,760 cytosine-phosphate-guanine (CpG) sites associated with smoking at epigenome-wide significance [5]. Annotation of these CpG sites to genes pointed towards differential methylation for almost one-third of all genes in the human genome, indicating an extensive effect of smoking on DNAm levels.

Findings from blood have been partially replicated in other tissues such as the lungs [6,7] and adipose tissue [8]. As the brain is involved in the development and maintenance of tobacco use disorder (TUD), DNAm changes in the brain are of interest but knowledge on smoking-associated DNAm signatures remains sparse. One EWAS has examined DNAm signatures of smoking in the nucleus accumbens (NAc, *n* = 221) using postmortem tissue, finding seven CpG sites to be associated with smoking at epigenome-wide significance [9]. None of these were found to be significant in an EWAS of smoking in blood, which suggests heterogeneity of associations between tissues [9]. Deciphering these tissue-specific and tissue-shared methylation patterns could provide insights into a potential proxy function of blood for predicting differential methylation in the brain.

In substance use disorders (SUDs), a whole neurocircuitry of addiction consisting of multiple cortical, striatal, and limbic brain regions exhibits functional changes [10]. Cortical regions such as the prefrontal and the anterior cingulate cortex (ACC) are involved in executive control and are especially important in the preoccupation/anticipation stage of the addiction cycle [10]. The striatum is subdivided into a ventral (VS) and a dorsal part with the latter comprising the caudate nucleus (CN) and the putamen (PUT). The VS is thought to be related to reward processing [11], whereas CN and PUT are involved in sensorimotor processing and the habitual behavior observed in later stages of addiction [12].

To investigate differential methylation in the context of smoking, methylation changes within brain regions which are part of the neurocircuitry of addiction need to be assessed. In the present study, we investigated the tissue-specificity of smoking-associated methylation signatures by performing EWASs of smoking and evaluating the similarity of smoking-associated methylation patterns between blood and multiple brain regions (ACC, Brodmann Area 9 (BA9), CN, PUT and the VS). Based on the results, we performed downstream analyses including the assessment of differentially methylated regions (DMRs), gene ontology (GO) enrichment analysis, and GWAS enrichment analysis.

## 2. Materials and Methods

### 2.1. Samples

Postmortem human brain tissue was obtained from the New South Wales Tissue Resource Center (NSWTRC, University of Sydney, Sydney, Australia), as part of a previous study on alcohol use disorder (AUD) [13]. Information on smoking was available for a total of 304 postmortem brain samples originating from 80 European American tissue donors. For each donor, tissue from at least one of the five brain regions (ACC, BA9, CN, PUT, and VS) was available. BA9 brain samples from an additional 12 subjects were obtained from the University of Texas Health Science Center at Houston (UTHealth, Prof. Consuelo Walss-Bass). Here, sample donors were of Asian (*n* = 1), Black (*n* = 1), Hispanic (*n* = 2), and White (*n* = 8) ethnicities. For 10 of the 12 subjects, matched blood samples were available. Total sample sizes of different brain regions ranged from 38 to 72 (Appendix A). Inclusion criteria were age > 18 and no history of severe psychiatric, neurodevelopmental, or substance use disorder (except AUD and TUD). These criteria as well as smoking status were assessed in next-of-kin interviews. Subjects included in the present study had a smoking status of either current smoking or never smoking prior to death. A descriptive summary of phenotypic information on tissue donors is shown in Table 1.

### 2.2. DNA Extraction, DNAm Analysis, and Quality Control

Extraction of DNA, sample randomization, epigenome-wide DNAm analysis, and quality control of methylation data was performed as described in Zillich et al. [13]. In brief, epigenome-wide DNAm was determined using the Infinium Human Methylation EPIC BeadChip (Illumina, San Diego, CA, USA) and quality control was performed with a customized and updated version of the CPACOR pipeline [14]. Samples from the NSWTRC were processed separately for each brain region in either a single or in two batches (ACC and PUT: one batch, BA9, CN and VS: two batches). The UTHealth samples were processed in a single batch after randomization according to sex, AUD status, and tissue. Methylation data were quantile-normalized.

### 2.3. Statistical Analyses

All statistical analyses were performed using R version 3.6.1 [15]. A schematic workflow depicting the different analysis levels within the present study is shown in Figure 1.

#### 2.3.1. Between-Tissue Correlation

M-values of methylation were generated by logit-transformation of β-values as described by Du, Zhang [16]. For each of the 632,086 CpG sites remaining after QC, average M-values were calculated separately in blood and brain samples. Correlation of mean M-values between blood and brain (*n* = 10) was determined using the Pearson correlation method.

#### 2.3.2. EWAS in Five Brain Regions

Separate EWASs were performed to identify smoking associated CpG sites in each of the five brain regions (ACC, BA9, CN, PUT, VS). The EWASs were restricted to samples originating from donors with European American ancestry to reduce confounding by ancestry. A linear regression model with M-values specified as the dependent variable and smoking status as a predictor was used. As covariates, sex, age, postmortem interval (PMI), AUD status, batch, and the first 10 principal components of control probes (PCcp) were included. Neuronal cell fractions were estimated using the Houseman approach [17] with the dorsolateral prefrontal cortex reference dataset [18], and included as a covariate. For BA9, tissue from both resources (NSWTRC and UTHealth) was combined in the EWAS, whereas for all other brain regions, tissue samples from NSWTRC were used exclusively. In the EWAS of smoking in BA9, the first four genotype principal components (PCgeno) were used as additional covariates to correct for genotype differences between sample donors. Genotype data of NSWTRC samples were generated using the Illumina Infinium OMNI5 array (Illumina, San Diego, CA, USA). The Illumina Infinium Global Screening Array (Illumina, San Diego, CA, USA) was used for genotyping of UTHealth samples. Genotype QC was performed as described by Turner, Armstrong [19]. Sample sizes used in the EWAS were *n* = 38 (ACC), *n* = 65 (VS), *n* = 68 (CN and PUT), and *n* = 72 (BA9). Prior to EWAS, a variance inflation factor (VIF) analysis was performed to investigate multicollinearity in the linear model. If a VIF larger than 10 was detected, one of the correlated covariates was removed in a way so that the final models for each brain region were most comparable to each other (Appendix A). *p*-values obtained in the association analyses were FDR-adjusted for multiple testing using the Benjamini-Hochberg procedure [20].

#### 2.3.3. Differentially Methylated Regions

Differentially methylated regions were identified using the comb-p software (v. 0.50.2) [21]. The following DMR definition was used: seed *p*-values < 0.01 for a minimum number of two CpG sites within a 500 bp genomic window. Correction for multiple testing was performed using the Sĭdaák method as implemented in comb-p.

#### 2.3.4. Gene Ontology (GO) Enrichment Analysis

Functional enrichment of genes harboring smoking-associated CpG sites within GO terms was investigated using missMethyl [22] (v. 1.20.4). Probes were annotated to genes according to the Illumina EPIC manifest file (http://webdata.illumina.com.s3-website-us-east-1.amazonaws.com/downloads/productfiles/methylationEPIC/infinium-methylationepic-v-1-0-b4-manifest-file-csv.zip) as downloaded on 10 August 2018. Analysis was restricted to CpG sites with association *p*-values < 0.001.

#### 2.3.5. GWAS Enrichment Analysis

To investigate a potential overlap between polygenic and polyepigenetic signal, we performed GWAS enrichment analysis. Gene sets were derived from EWAS results for each of the five brain regions. For this, CpG sites exhibiting a nominally significant (*p* < 0.05) association with smoking were annotated to genes. Enrichment of these gene sets within GWAS signals of different smoking phenotypes and SUDs was investigated using Multi-marker Analysis of GenoMic Annotation (MAGMA, v. 1.08b) [23]. Summary statistics for smoking phenotypes were obtained from Liu [24]: (1) smoking initiation (ever/never, median n/SNP = 632,802), (2) age of initiation (median n/SNP = 262,990), (3) smoking cessation (median n/SNP = 312,821), and (4) cigarettes per day (median n/SNP = 263,954). For SUDs, GWASs on AUD (*n* = 313,959) [25], opioid use disorder (OUD, dependent vs. unexposed, *n* = 37,003) [26], and cannabis use disorder (CUD, *n* = 374,287) [27] were tested. Summary statistics for problematic alcohol use (PAU, *n* = 435,563) [25] were also included. Significance level was adjusted to Bonferroni correction for multiple testing of the 8 GWASs (*n* = 8, *p* < 6.25 × 10^−3^).

## 3. Results

While sample sizes ranged between 38 and 72 per brain region, tissue of 92 donors was included in the present study. Sample characteristics are displayed in Table 1 and a detailed breakdown of batches and brain regions is available in Appendix A, along with the causes of death in Appendix A.

### 3.1. Correlation of Methylation Levels

Hierarchical clustering based on M-values of all CpG sites (*n* = 632,086) revealed a clear distinction based on tissue of origin (Figure 2A). At the same time, the between-tissue correlation between averaged methylation levels of blood and brain was high, with a Pearson correlation of r = 0.91, *p* < 2.2 × 10^−16^ (Figure 2B).

### 3.2. Epigenome-Wide Association Studies

To expand the analysis of methylation signatures of smoking to the neurocircuitry of addiction, we performed EWASs of smoking in five brain regions. In the ACC, one CpG site (cg152925993) was differentially methylated at epigenome-wide significance between smokers and non-smokers (b = 0.53, se = 0.07, *p* = 7.33 × 10^−8^, FDR = 0.048, *n* = 38). This CpG site was annotated to the long non-coding RNA (lncRNA) *LOC339975*. No epigenome-wide significant CpG sites were identified in the other brain regions. The strongest association in BA9 was observed for cg22909901 (b = 0.34, se = 0.06, *p* = 9.99 × 10^−7^, FDR = 0.652, *n* = 72) which was annotated to *MCF2L2/B3GNT5*. In the ventral striatum, cg09139806 in *RNF220* was most significant (b = −0.534, se = 0.10, *p* = 1.15 × 10^−6^, FDR = 0.364, *n* = 65). In the dorsal striatum, the strongest associations with smoking were observed for cg07022048 in *KRT7* for the caudate nucleus (b = −0.22, se = 0.04, *p* = 2.66 × 10^−7^, FDR = 0.173, *n* = 68) and for cg18712580 in *STK38L* (b = −0.64, se = 0.12, *p* = 1.87 × 10^−6^, FDR = 1, *n* = 68) in the putamen. A summary table including EWAS sample sizes, genomic inflation factors and resulting FDR-thresholds is provided in Appendix A. Manhattan plots for ACC, BA9, and VS are provided in Figure 3 and for CN and PUT in Appendix A. Appendix A summarize the top 10 associations from each brain region.

### 3.3. Differentially Methylated Regions

We identified a total of 16 DMRs in the five brain regions that were associated with tobacco smoking. Two DMRs were observed in multiple brain regions. One was a region consisting of 5 probes in *PRSS50* in ACC and VS, and another contained the *LINC00612/A2M-AS1* genes in both dorsal striatal regions (CN and PUT). DMRs for each brain region are summarized in Table 2 and are highlighted using green gene names in the Manhattan plots.

### 3.4. Gene-Ontology Analysis

In the ACC, smoking-associated CpG sites were enriched for GO terms related to neurodevelopment, cell growth, and morphogenesis. Enriched GO terms in BA9 were related to dendritic spine development and chromatin modification. In the VS, neuron-specific pathways were enriched, as well as processes associated with the regulation of vessel development. In both dorsal striatal regions, genes harboring differentially methylated CpG sites were enriched for GO terms related to immune pathways. After correction for multiple testing, no GO terms remained statistically significant. Result tables for the top 10 associated GO terms are shown in Appendix A.

### 3.5. GWAS Enrichment Analysis

Gene sets consisting of smoking-associated CpG sites were overrepresented in GWAS summary statistics of smoking traits, such as smoking initiation, age of initiation, and cigarettes per day. Furthermore, significant enrichment for other SUDs, such as cannabis-, alcohol-, and opioid use disorder was observed. Statistical significance of enrichment for all tested gene-sets is displayed in Figure 4.

### 3.6. Consistency of Smoking-Associated CpG Sites in EWAS Results of Blood and Brain

We examined differential methylation of CpG sites in the brain, which have previously been suggested to predict smoking status in peripheral blood. When we investigated the nine available CpG sites from the prediction model by Maas et al. [28], we found no consistent differential methylation across the five brain regions. For two of them, nominally significant associations were detected. The *AHRR* CpG site cg05575921, even used as a single-marker smoking status predictor, was significantly associated with smoking in the ACC (*p* = 0.038) and PUT (*p* = 0.033). cg21566642, an intergenic CpG site, was associated with smoking in BA9 (*p* = 0.018). Full results of the consistency analysis are listed in Table 3.

## 4. Discussion

In the present study, we investigated the consistency of smoking-associated methylation signatures in blood and brain, examined differential methylation in the brain associated with smoking status, and performed several EWAS downstream analyses in five brain regions related to the neurocircuitry of addiction. The strong overall correlation of methylation levels in matched blood and brain samples is in line with previous findings of high cross-tissue correlation coefficients [29,30].

For the EWAS of smoking in the ACC, one epigenome-wide significant CpG site was observed within *LOC339975*. We identified a total of 16 differentially methylated regions associated with smoking status. A DMR in *PRSS50* was shared between the ACC and the VS. A functional role of PRSS50 in the brain has not been systematically evaluated so far. However, in cancer cells, knockdown of *PRSS50* resulted in impaired cell proliferation and increased levels of apoptosis [31]. Also, promoter hypermethylation of *PRSS50* was detected in an EWAS of age-related macular degeneration (AMD) in blood and retinal tissue [32]. In the present study, also hypermethylation was observed for the *PRSS50* DMR in the ACC and the VS. The second smoking-associated DMR shared between two regions of the brain (CN and PUT) was annotated to *LINC00612/A2M-AS1*. Both lncRNAs, *LINC00612* and *A2M-AS1*, are involved in inflammatory processes. An anti-inflammatory function disrupted by smoking has been discovered for *LINC00612* in the lungs [33], while *A2M-AS1* has been linked to interleukin 1 receptor signaling in cardiomyocytes [34]. Further studies need to investigate if these lncRNAs are also involved in inflammatory signaling in the brain. Functional enrichment of inflammatory and immune-related processes was also supported by the results of the GO enrichment analysis: in PUT, genes harboring smoking-associated CpG sites were enriched in immune cell migration pathways. In the ACC, BA9, and the VS, GO enrichment analysis revealed pathway enrichment related to neuronal development and morphogenesis. Given the direct influence of nicotine on immune cell function [35] and neuronal development [36], results from the present study may point towards an additional epigenetic effect of smoking on these cellular processes.

GWAS enrichment analysis revealed significant overrepresentation of EWAS-derived gene-sets within GWAS signals of several smoking phenotypes and SUDs. This points towards a fraction of genes detected by GWAS and EWAS contributing to the development and maintenance of tobacco smoking and SUDs. Further research needs to uncover how genetic and epigenetic mechanisms collectively contribute to the disease course in SUDs.

The well-known *AHRR* CpG site cg05575921 was associated with smoking in the ACC and PUT and was consistently hypomethylated in all investigated brain regions. As hypomethylation and significant association with smoking has previously also been detected for cg05575921 in blood [5], the lungs [6] and in adipose tissue [8], it may be a common locus representing smoking status across tissues.

In contrast to the strong general correlation of methylation levels between tissues, associations with smoking were heterogenous between tissues. Blood samples might thus be limited in the extent to which they can function as a proxy for smoking-associated differential methylation in the brain. Concurrently, a potential specificity of associations implies that specific methylation signatures of smoking in the brain could contribute to the pathophysiology of TUD. This is also supported by the overlap of polygenic and poly-epigenetic signals identified for smoking phenotypes. Nevertheless, smoking could also have a non-specific effect on the entire organism and further studies are needed to investigate the functional relevance of smoking-associated differential methylation within the neurocircuitry of addiction and its contribution to the development and maintenance of TUD.

Several limitations need to be addressed. First, a large fraction of tissue donors was diagnosed with AUD prior to death. Despite adjustment for AUD in the linear model, we cannot rule out residual confounding by alcohol consumption and future studies should investigate smoking independent of other SUDs. However, a recent Mendelian randomization study has shown that a smoking-specific risk on other disorders remains after correcting for the genetic risk for alcohol consumption [37]. Second, smoking status was assessed based on next-of-kin reports which are known to be less precise than direct measurement of the nicotine metabolite cotinine [38]. Smoking status and TUD status often overlap, but TUD was not specifically assessed in the present study. A recent EWAS of smoking status and TUD found both specific and shared methylation signals between smoking status and lifetime TUD [39], which underlines the importance of investigating both traits simultaneously. Third, due to multiple testing, even with our largest sample size of *N* = 72, the EWAS of smoking in the brain is still insufficiently powered. At the same time, postmortem brain tissue is scarce which makes it challenging to obtain enough samples.

The present study identified smoking-associated DNAm changes in the neurocircuitry of addiction related to immunological and neurodevelopmental processes. However, certain DNAm signatures might represent a predisposition to tobacco smoking rather than a consequence of it. Follow-up studies should thus investigate tissues of donors deceased at different stages of TUD, which may enable the identification of changes in DNAm levels during the disease course and differentiate between predispositions and consequences of smoking. The functional consequences of smoking-associated changes in DNAm levels should also be addressed in a more comprehensive design, for example using a multi-omics approach integrating methylation and transcriptomic data. Ultimately, deeper insight into methylation changes within the neurocircuitry of addiction could lay the foundation for better understanding of the pathomechanisms of tobacco use disorder.

## Figures and Tables

**Figure 1 jpm-12-00566-f001:**
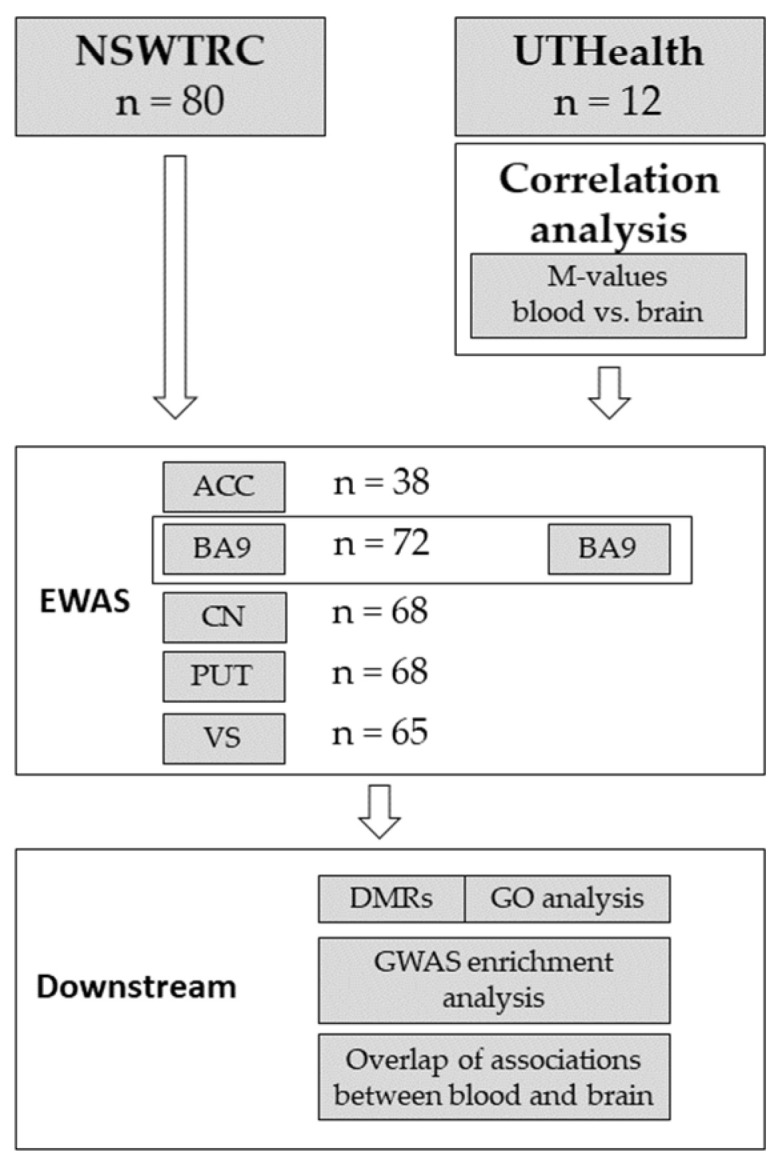
Analysis flowchart.

**Figure 2 jpm-12-00566-f002:**
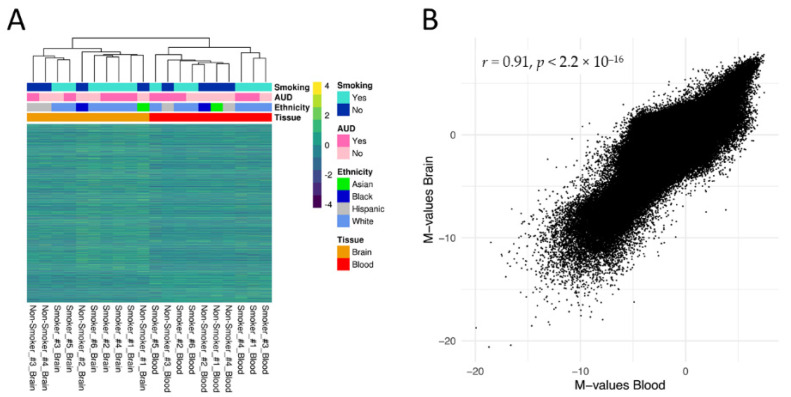
(**A**) Hierarchical clustering of matched blood and Brodmann Area 9 (BA9) brain samples (*n* = 10) based on methylation M-values (row normalized) of the *n* = 632,086 CpG sites remaining after quality control. Next to the tissue type, smoking status, alcohol use disorder status and ethnicity are displayed as additional phenotypic variables. (**B**) Pearson correlation of mean M-values in the between-subject correlation analysis using matched blood and BA9 brain samples.

**Figure 3 jpm-12-00566-f003:**
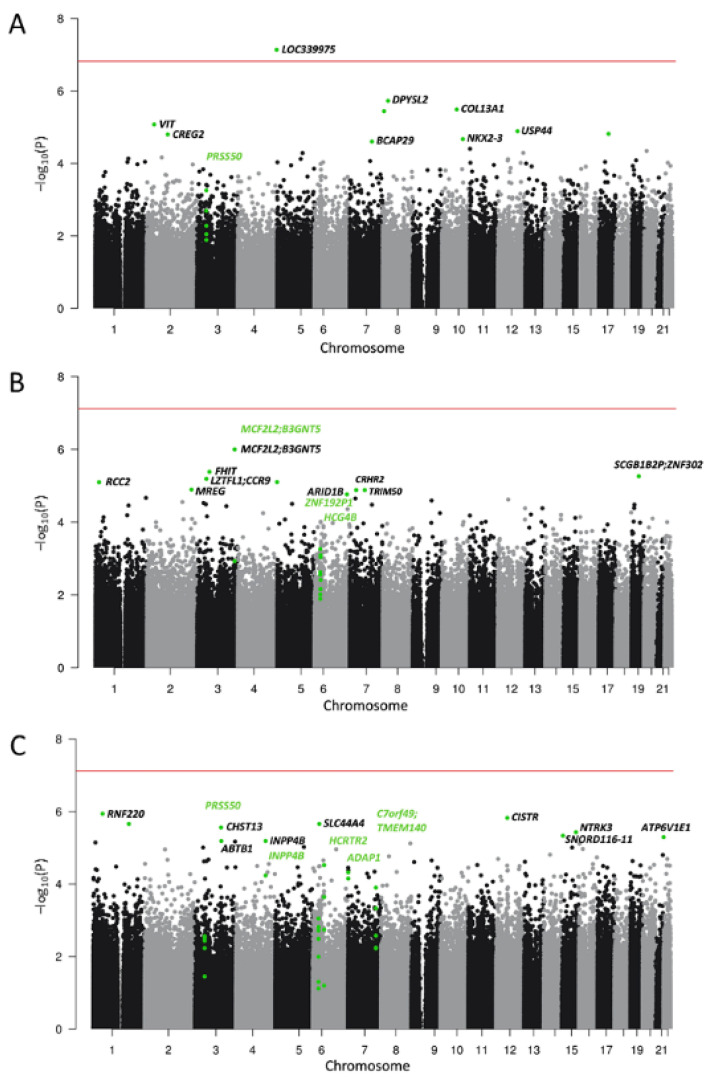
Manhattan plots for the EWAS of smoking in (**A**) anterior cingulate cortex (ACC), (**B**) Brodmann Area 9 (BA9), and (**C**) the ventral striatum (VS). Top 10 smoking-associated CpG sites for each region and CpG sites contributing to differentially methylated regions (DMRs) are highlighted in green. Annotation to gene names was performed where possible. Names of genes harboring a DMR are highlighted in green. The horizontal line depicts epigenome-wide significance (5% FDR, for further information, see Appendix A).

**Figure 4 jpm-12-00566-f004:**
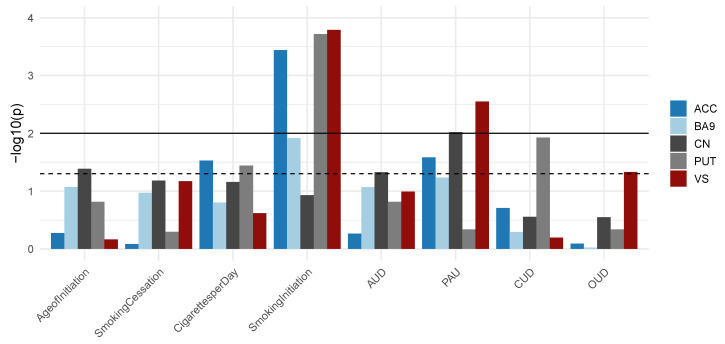
Summary of the results from GWAS enrichment analysis. For each brain region and each GWAS, the statistical significance of enrichment is shown. Dashed line represents nominal significance (*p* < 5 × 10^−2^) while the solid line depicts the significance threshold after Bonferroni correction for multiple testing (*n* = 8, *p* < 6.25 × 10^−3^). AUD, alcohol use disorder; PAU, problematic alcohol use; CUD, cannabis use disorder; OUD, opioid use disorder.

**Table 1 jpm-12-00566-t001:** Demographic data of tissue donors.

	NSWTRC			UTHealth		
Characteristic	Smokers	Non-Smokers	*p*	Smokers	Non-Smokers	*p*
*n*	50	30		7	5	
Age (years)	56.14 (9.00)	57.83 (10.50)	0.47	54.86 (9.48)	50.20 (10.80)	0.46
Sex (M/F)	40/10	18/12	0.09	6/1	5/0	1.00
PMI (hours)	32.17 (15.22)	34.20 (18.09)	0.61	31.00 (10.96)	33.62 (9.69)	0.67
AUD (% cases)	72.00	20.00	1.89 × 10^−5^	71.43	40.00	0.62

For quantitative measures, standard deviation (±SD) is provided in brackets. NSWTRC, New South Wales Tissue Research Center; UTHealth, University of Texas Health Science Center at Houston; PMI, postmortem interval; AUD, alcohol use disorder, *p*: *p*-value of a *t*-Test/Chi-squared test assessing the significance of differences between smokers and non-smokers.

**Table 2 jpm-12-00566-t002:** Differentially methylated regions (DMRs) identified in the five brain regions.

	Chr	CpG	Gene	Dir ^1^	N Probes	Z P	Z Sidak p	Start	End
ACC	3	cg02834909	* **PRSS50** *	+	5	1.72 × 10^−8^	7.48 × 10^−5^	46,759,438	46,759,589
BA9	3	cg22909901	*MCF2L2*; *B3GNT5*	+	2	3.71 × 10^−8^	3.19 × 10^−4^	182,981,632	182,981,708
	6	cg23161317	*ZNF192P1*	−	6	3.17 × 10^−9^	1.20 × 10^−5^	28,129,313	28,129,486
	6	cg23681866	*HCG4B*	−	5	1.48 × 10^−9^	6.66 × 10^−6^	29,895,116	29,895,261
CN	1	cg10703826	*TBX15*	−	7	5.96 × 10^−10^	2.03 × 10^−6^	119,532,044	119,532,234
	12	cg26114124	***LINC00612***; ***A2M-AS1***	−	8	3.02 × 10^−12^	5.60 × 10^−9^	9,217,510	9,217,860
	16	cg06751612	*PIEZO1*; *LOC100289580*	+	2	1.96 × 10^−7^	4.38 × 10^−3^	88,798,826	88,798,855
	5	cg11916729	*SCGB3A1*	+	6	2.81 × 10^−7^	1.82 × 10^−3^	180,018,465	180,018,565
PUT	12	cg02883147	***LINC00612***; ***A2M-AS1***	−	5	9.86 × 10^−9^	3.58 × 10^−^^5^	9,217,669	9,217,860
	19	cg24716275	*ZNF264*	−	6	7.80 × 10^−10^	4.17 × 10^−6^	57,702,371	57,702,501
VS	3	cg00817731	* **PRSS50** *	+	5	1.71 × 10^−7^	7.52 × 10^−4^	46,759,438	46,759,589
	4	cg08064687	*INPP4B*	−	2	7.43 × 10^−9^	8.65 × 10^−5^	143,326,485	143,326,542
	6	cg14654363	*-*	−	8	1.00 × 10^−9^	4.28 × 10^−6^	28,601,365	28,601,520
	6	cg27596495	*HCRTR2*	+	4	1.37 × 10^−8^	6.00 × 10^−5^	55,039,232	55,039,383
	7	cg06036947	*ADAP1*	−	2	5.34 × 10^−8^	4.60 × 10^−4^	949,758	949,835
	7	cg07972322	*C7orf49*; *TMEM140*	−	5	4.84 × 10^−10^	1.23 × 10^−6^	134,832,770	134,833,032

DMRs associated with tobacco smoking in the anterior cingulate cortex (ACC), Brodmann Area 9 (BA9), caudate nucleus (CN), putamen (PUT) and ventral striatum (VS). Genes exhibiting DMRs within multiple brain regions are highlighted in bold type. Dir ^1^: DMR characterized by hypermethylation (+) or hypomethylation (−) of probes.

**Table 3 jpm-12-00566-t003:** Associations of the predictor CpG sites in blood and brain.

CpG Site	Gene	Blood * (*n* = 15,907)		ACC (*n* = 38)		BA9 (*n* = 72)		CN (*n* = 68)		PUT (*n* = 68)		VS (*n* = 65)	
		*β*	*pval*	*β*	*pval*	*β*	*pval*	*β*	*pval*	*β*	*pval*	*β*	*pval*
cg05575921	*AHRR*	−0.180	4.55 × 10^−26^	−0.310	0.038	−0.158	0.163	−0.247	0.059	−0.220	0.033	−0.030	0.827
cg03636183	*F2RL3*	−0.095	1.12 × 10^−20^	−0.117	0.118	−0.010	0.834	−0.056	0.198	−0.016	0.619	−0.080	0.116
cg09935388	*GFI1*	−0.083	3.14 × 10^−17^	−0.162	0.345	−0.183	0.132	0.163	0.206	−0.060	0.595	0.142	0.369
cg12876356	*GFI1*	−0.057	1.49 × 10^−15^	0.022	0.838	−0.041	0.516	0.028	0.640	0.025	0.705	0.028	0.730
cg12803068	*MYO1G*	0.063	9.06 × 10^−23^	0.075	0.556	−0.045	0.598	0.101	0.163	0.082	0.292	0.014	0.810
cg13039251	*PDZD2*	0.030	1.36 × 10^−15^	0.060	0.673	−0.051	0.519	0.109	0.217	0.118	0.186	−0.014	0.894
cg01940273	*-*	−0.082	2.03 × 10^−34^	−0.015	0.895	−0.035	0.569	−0.008	0.903	0.007	0.885	−0.013	0.851
cg15693572	*-*	0.053	3.85 × 10^−11^	−0.120	0.297	0.068	0.478	0.113	0.227	−0.009	0.912	−0.030	0.800
cg21566642	*-*	−0.126	4.22 × 10^−25^	−0.205	0.202	−0.176	**0.018**	0.039	0.707	−0.147	0.119	0.048	0.628

For the nine CpG sites derived from Maas, Vidaki [28], effect sizes and ***p***-values of the association analyses with smoking in blood (* EWAS from Joehanes, Just [5]) and the five brain regions are summarized. Nominally significant results are highlighted in bold type. Β, EWAS effect size; pval, EWAS ***p***-value; ACC, anterior cingulate cortex; BA9, Brodmann Area 9; CN, caudate nucleus; PUT, putamen; VS, ventral striatum.

## Data Availability

Raw data and summary statistics for all analyses are available on request.

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
