# Peer review of "Epigenetic Signatures of Smoking in Five Brain Regions"

_jpm, 2022, doi:10.3390/jpm12040566_

Round 1
Reviewer 1 Report
To Authors:
The authors investigated the similarity of DNAm signatures between tissues, analyzing matched blood and post-mortem brain samples, and performed EWASs in five brain regions contributing to the neurocircuitry of addiction, resulting in the outcomes that cg15925993 within the LOC339975 gene was epigenome-wide significant in the ACC and functional enrichment was detected for biological processes related to neuronal development, inflammatory signaling and immune cell migration. The manuscript is largely well written and informative overall. However, there seem to be several major and minor concerns in this manuscript. The paper will be improved when the authors revise them according to the following comments:
[Major point]
To investigate specific CpG sites associated with smoking in the brain, the authors performed EWASs in five brain regions contributing to the neurocircuitry of addiction. Although they performed EWASs for smoking status of either current smoking or never smoking prior to death, it would also be noteworthy to examine differential DNA-methylation between cases with TUD and controls (i.e., never smokers), just as they examined differential DNA-methylation between cases with severe AUD and controls in their previous study [13]. If they are not planning to submit another paper about EWASs for TUD status and not smoking status elsewhere, it would be better for the authors to present some data about EWASs for TUD status even as supplementary files as well as the data about EWASs for smoking status if possible, or at least add some description about such analyses related to TUD status in the Introduction or Discussion section if not possible.
[Minor points]
- Materials and Methods:
Although the present study was conducted by utilizing the postmortem human brain tissue obtained from the New South Wales Tissue Resource Center (NSWTRC, University of Sydney, Australia) as part of a previous study on alcohol use disorder (AUD) [13], the authors should describe the methods in more details (e.g., which Beadchip was used, how each analysis was conducted to compare what data, etc.)
Table 1:
Although the information of AUD patients (% cases) are presented in the table, the information of TUD patients (% cases) are not presented. It would be better to add the information of TUD patients (% cases) among smokers.
Reviewer 2 Report
Major comments
1. The authors sought “to evaluate differential methylation in the context of smoking, methylation changes within individual brain regions contributing to the neurocircuitry of addiction.” How do these findings contribute to the neurocircuitry of addiction? Do the authors mean to say to the neurobiology of addiction instead?
2. The discussion section provided more results instead of arguing on the scientific meanings of the findings?
The sample comprises individuals with tobacco and other substance use disorders. Was there is any possible comparison with individuals with no history of SUDs? If not, this can be discussed as a limitation.
3. How are these results fit with Mendelian randomization studies of smoking in GWAS from several ancestries?
4-Are methylation changes from tobacco perhaps everywhere and nonspecific to the brain? Thus, the changes may be very far from a possible explanation of the neurobiology of smoking as an addictive disorder.
Minor comments
To evaluate whether the methylation signatures of smoking in blood serve as a proxy for findings in the brain, we 23 investigated the similarity of DNAm signatures between tissues, analyzing matched blood and postmortem brain samples (N = 10). To investigate specific CpG sites associated with smoking in the brain, we performed EWASs in five brains.
This sentence can be rewritten for clarity. To evaluate… and to investigate. Is it to evaluate, authors investigated? Overall, the abstract needs to be more structured and easier to grasp.
Round 2
Reviewer 1 Report
To Authors:
The authors revised the manuscript according to my previous comments. The manuscript is largely well written and informative overall, so there seem to be no concerns in this manuscript. The paper may be accepted in the present form.